# We Won’t Go There: Barriers to Accessing Maternal and Newborn Care in District Thatta, Pakistan

**DOI:** 10.3390/healthcare9101314

**Published:** 2021-10-01

**Authors:** Muhammad Asim, Sarah Saleem, Zarak Husain Ahmed, Imran Naeem, Farina Abrejo, Zafar Fatmi, Sameen Siddiqi

**Affiliations:** Department of Community Health Sciences, Aga Khan University, Karachi 74800, Pakistan; sarah.saleem@aku.edu (S.S.); zarak.ahmed@aku.edu (Z.H.A.); imran.naeem@aku.edu (I.N.); farina.abrejo@aku.edu (F.A.); zafar.fatmi@aku.edu (Z.F.); sameen.siddiqi@aku.edu (S.S.)

**Keywords:** health-seeking behavior, healthcare, service utilization, maternal and newborn care

## Abstract

Accessibility and utilization of healthcare plays a significant role in preventing complications during pregnancy, labor, and the early postnatal period. However, multiple barriers can prevent women from accessing services. The aim of this study was to explore the multifaceted barriers that inhibit women from seeking maternal and newborn health care in Thatta, Sindh, Pakistan. This study employed an interpretive research design using a purposive sampling approach. Pre-tested, semi-structured interview guides were used for data collection. The data were collected through eight focus group discussions with men and women, and six in-depth interviews with lady health workers and analyzed through thematic analysis. The study identified individual, sociocultural, and structural-level barriers that inhibit women from seeking maternal and newborn care. Individual barriers included mistrust towards public health facilities and inadequate symptom recognition. The three identified sociocultural barriers were aversion to biomedical interventions, gendered imbalances in decision making, and women’s restricted mobility. The structural barriers included ineffective referral systems and prohibitively expensive transportation services. Increasing the coverage of healthcare service without addressing the multifaceted barriers that influence service utilization will not reduce the burden of maternal and neonatal mortality. As this study reveals, care seeking is influenced by a diverse array of barriers that are individual, sociocultural, and structural in nature. A combination of capacity development, health awareness, and structural interventions can address many if not all of these barriers.

## 1. Introduction

Globally, about 85% of obstetric complications occur during labor and the early postnatal period [1]. The World Health Organization points out that nearly 75% of maternal deaths caused by pre-eclampsia, sepsis, severe bleeding, unsafe abortion, and complications in childbirth can be prevented by providing high quality antenatal care, skilled attendance at birth, and timely referrals [2]. Together, these interventions make part of the World Health Organization’s (WHO) birth preparedness and complication readiness strategy for low and middle income countries (LMIC) [3]. Despite their proven efficacy, these interventions are yet to be successfully implemented in Pakistan, a country with the third highest burden of maternal, fetal, and child mortality in the world [4]. For instance, research from the country indicates that antenatal care is inadequate in coverage and quality; it is estimated that only 51% women receive antenatal care in Pakistan [5]. Furthermore, the content of antenatal counseling fails to include numerous important facets relating to maternal and newborn health [6]. According to a national survey, only 45%, 47%, and 64% women reported receiving counseling on the early initiation of breastfeeding, exclusive breastfeeding, and nutritional needs during pregnancy, respectively [5]. In addition, the same survey also reported that only 62% women in Pakistan delivered their baby in the presence of a skilled provider [5]. 

Health services utilization is particularly poor in rural areas of Pakistan where 41% of deliveries take place at the home and 38% are assisted by unskilled birth attendants [5]. Moreover, pregnant women in rural Pakistan ignore obstetric care, [7] avoid consuming micronutrients, [8] and refuse vaccination against neonatal tetanus [9,10]. These factors can explain why maternal and neonatal health indicators are significantly worse in rural areas of the country. The prime example of this comes from the district of Thatta, where the maternal (313/100,000 live births), neonatal (50/1000 live births), and perinatal (95.2/1000 births) mortality figures are the highest across low and middle income countries [11,12]. This trend is occurring despite concerted efforts by the government to improve the quality, coverage, and demand for health care services in rural Pakistan through the formation of public–private partnerships [13]. 

Research from LMICs indicates that poor health service utilization can be related to the presence of certain cultural beliefs which impact maternal and newborn health seeking behavior [14,15,16]. Studies from African and South Asian countries show that traditional belief systems, religious beliefs, and gender power dynamics play a role in hindering women from accessing and utilizing services [17]. These behaviors can lead to poor compliance of antenatal and postnatal care [18], poor nutritional habits [19], home deliveries [20], lack of thermal care, and delayed initiation of breastfeeding [21]. 

Despite this mounting evidence, no study to date has conducted an in-depth exploration of the multi-faceted barriers to maternal and newborn health service utilization in rural Pakistan. Previous studies have determined the rates of service utilization without exploring the sociocultural determinants of health-seeking behavior [22,23,24]. Addressing this gap, we designed this interpretive study to interact with both community members and health care providers to identify the multifaceted barriers to health service utilization in rural Pakistan. 

## 2. Methods

This study was conducted in Thatta, a rural district in the southern province of Pakistan which is categorized in the low human development index strata [25]. Only 17% of the women living in the district are literate, and 40% of the births take place at home [12]. We purposively selected this district as it reports the highest maternal and neonatal mortality across low- and middle-income countries 11,12. Considering the dearth of literature regarding barriers to accessibility and the utilization of maternal and newborn healthcare in rural Pakistan, we employed an interpretive research methodology. Through this process, we utilized in-depth interviews and focus group discussions (FGDs) to explore barriers that inhibit care seeking during pregnancy.

### 2.1. Study Participants

To obtain a broad perspective, we included three categories of respondents: married men, married women of reproductive age, and lady health workers (LHWs) from lady health workers program [26]. The criteria for recruiting men and women stipulated that they have at least one child under the age of two years. This ensured a strong recall of their pregnancy experience. In addition, we selected LHWs with a minimum of five years of experience working within the district. 

### 2.2. Interview Guides Development

A semi-structured guide was initially developed for our in-depth interviews with LHWs. This guide was formulated after an extensive literature review using different keywords related to the determinants of health service utilization in LMICs. Following data collection with LHWs, we used the results to develop two further semi-structured interview guides for FGDs with men and women. All interview guides were pilot tested in an adjacent district before data collection and were updated periodically as we learned more about the community. The major sections of the guides related to: perception of health facilities, health seeking practices, and social constraints. These themes had several probing statements to further explore health service utilization. 

### 2.3. Data Collection

Data were collected from March to July 2019. In total, we conducted six in-depth interviews with LHWs, four FGDs with men, and four FGDs with women (see Table 1). During the first phase of the study, we conducted the in-depth interviews with lady health workers. The interviews were conducted face-to-face in Urdu at the local health facility by the first author (MA). After completing the in-depth interviews, we used the results to refine our semi-structured interview guide for FGDs. We then commenced the second phase of data collection, which included male and female FGDs. All FGDs were conducted at a convenient location mutually agreed on by all participants. Each FGD session hosted 6–7 participants and was moderated by the first author (MA) with the help of two research assistants.

In-depth interviews lasted between 20 and 30 min, and focus group discussions lasted 40 to 50 min. All interviews and discussions were audio recorded and accompanied by written field notes. A debriefing session followed each interview and discussion to resolve any discrepancies in interpreting findings. Data collection was ceased upon reaching information saturation.

### 2.4. Ethical Consideration

The ethics review committee of Aga Khan University, Karachi, Pakistan [AKU-ERC-2020–0479-8902] approved the study protocols of the Rural Health Program of the Department of Community Health Sciences that enable us to design this study. Moreover, verbal informed consent was obtained from all the study participants before conducting interviews.

### 2.5. Analysis

All the recorded interviews were transcribed verbatim by the first author into the English language. The transcribed interviews were counter-checked with written notes by two research assistants to ensure the data quality of transcripts. The inductive method was used to formulate the study themes (see Figure 1); this approach refers to a detailed reading of raw data to derive concepts, themes, and interpretations of the participant’s responses [27]. A thematic analysis was carried out manually to analyze the data. A list of major themes was identified after a detailed reading of transcripts and field notes by the three co-authors (MA, ZHA, and SS). Interviews with healthcare providers, women and men, were analyzed simultaneously for greater understanding and to triangulate the study findings. Following this, data were filtered from the written notes to ‘meaning units’ and labeled with a ‘unique code’ without losing the study context and respondent’s identity. Codes were then analyzed and assembled into categories to capture the manifest meaning (Figure 1). We analyzed the data by utilizing the socioecological model and organized the themes into three broad categories: (1) individual, (2) sociocultural, and (3) structural. To ensure the authenticity of the findings, the data were triangulated by multiple data sources (healthcare providers, men, women, and field notes) and data collection methods (FGDs and IDIs) to compare alternative perspectives and minimize the chance of any misleading information. 

## 3. Results

Based on our analyses, we identified three broad categories of barriers that impacted the utilization of maternal and newborn healthcare services. These were conceptualized through the socioecological framework as individual, sociocultural, and structural-level barriers (Figure 1). These findings were divided into further subthemes and are presented below. The background characteristics of lady health workers are illustrated in Table 2. The sociodemographic characteristics of community participants are presented in Table 3.

### 3.1. Individual Barriers

Individual barriers refer to the personal beliefs and attitudes held by individuals that impact their ability to utilize health services. These included: mistrust towards public health facilities and inadequate symptom recognition. 

#### 3.1.1. Mistrust towards Public Health Facilities 

Both public and private health facilities are available within the district of Thatta. The public health facilities in the district include district hospitals, rural health centers, basic health units, and dispensaries. While many participants were keen to receive free treatment from these facilities, they did so only as a last resort owing to the apathetic attitude of doctors:

*“The doctors treat us like we are not humans. Imagine waiting for four hours and then the doctor only gives you 30 seconds of his time.”* (Men, 34 years, FGD)

Moreover, our interviews revealed that several participants were skeptical of the intentions of facility workers and felt that they were sabotaging their attempts to receive treatment. Highlighting this, one female respondent stated:

*“The staff is so rude and they do not care about patients I went …. for an ultrasound. The staff told me that the machine was not working, and I had to come again. Later, I found out that they are not operating the machine because the operator had decided to leave work early.”* (Woman, 29 years, FGD) 

Similarly, a male participant highlighted his views on the medication dispensed at public health facilities:

*“No matter what illness you go to the public hospital for they give you the same medication. Hypertension…diabetes…stomach pain it is the same medicine every time. What is worse is that these medicines don’t work. When we go to the private clinic, we get a different medicine and it always works.”* (Woman, 29 years, FGD)

These excerpts indicate that the residents of Thatta are reluctant to use public health facilities due to a general feeling of mistrust. It is likely that repeated experiences of poor treatment on part of doctors have helped fuel this notion of mistrust and hampered service utilization. 

#### 3.1.2. Inadequate Symptom Recognition 

Inadequate symptom recognition was identified as another individual level barrier to service utilization. In short, this refers to the inability for mothers to recognize complications in their early stages by paying limited attention to bodily signals. This is illustrated in the account provided below:

*“My face and hands were swollen for weeks and I was having headache and abdominal pain, but I thought it was nothing and just a normal part of pregnancy. It was only after my body starting shaking that I went to the hospital.”* (Woman, 24 years, FGD)

As a result of poor symptom recognition, women often neglect routine visits to the clinic and only seek the doctor in case of a severe emergency. Commenting on this, one LHW stated:

*“Women do not go to the clinic during pregnancy. They only go when something severe happens such as bleeding. They (pregnant women) think of abdominal pain and headaches as a routine part of life.”* (LHW, 48 years, FGD) 

These results indicate that the inability to recognize illness symptoms along with an attitude that positions doctors as last-minute saviors prevents service utilization in Thatta. 

### 3.2. Sociocultural Barriers

We conceptualize sociocultural barriers as certain rules and patterns of thought that stem from societal norms and values. While these constructs need not be universally held by all members of society, they exert their influence on health-seeking behavior and service utilization. In this study we identified three such barriers: aversion towards biomedical interventions, gendered imbalances in decision making, and restricted mobility for seeking care. 

#### 3.2.1. Aversion to Biomedical Interventions 

Many respondents held the view that all forms of biomedical interventions have certain side effects. They felt that while in the short run these interventions may alleviate symptoms, in the long run they would cause other complications. In contrast, they felt that home remedies were ideal as they would address the underlying problem without causing any side effects. This was illustrated by a male respondent who stated:

*“The local cure (Desi Ilaj) is always the best approach. If you take these medicines and injections, you will be worse off than you were. They (doctors) fix things in the short term. Using herbs such as Kalonji and Moringa are best.”* (Man, 48 years, FGD) 

As a possible result of this belief, our interviews revealed that pregnant women rarely take advantage of nutritional supplements such as folic acid, vitamins, and iron pills that are provided free of cost by the LHWs at the time of antenatal visits: 

*“During our door-to-door visits, we provide free folic acid and iron tablets. Pregnant women usually refuse because they think the micronutrients will cause pregnancy complications.”* (LHW, 37 years, IDI) 

When we questioned pregnant mothers on their reluctance to use vitamins and supplements, they highlighted that these pills would abnormally increase the size of their fetus and eventually cause a difficult delivery. According to one woman:

*“Vitamin pills increase the size of the fetus. Since they have started giving us these pills, we have to deliver our babies through cesarean. I cannot afford such a complicated delivery, I have no money.”* (Woman, 39 years, IDI) 

The aversion towards biomedical interventions was not limited to vitamins and supplements. Our interviews also revealed that mothers were hesitant towards being vaccinated against tetanus. In particular, they felt that the vaccine would result in miscarriages and stillbirths: 

*“We should not be vaccinating pregnant mothers. Their bodies cannot take what is in these injections. I will only get vaccinated in the 7th month of my pregnancy because it will cause an abortion in the first two trimesters.”* (Woman, 29 years, FGD)

To sum up, many commonly held practices such as dietary supplementation and vaccination that are considered essential to maternal and newborn health are not practiced in Thatta. Instead, pregnant mothers and their families show an apprehension towards biomedical interventions and associate them with negative consequences.

#### 3.2.2. Gendered Imbalances in Decision Making

Our research revealed gendered differences in selecting the place of delivery. For example, all the interviewed men preferred home deliveries, whereas most of the women aspired to deliver their babies at health facilities. Primarily, males preferred to have their children delivered at home for financial reasons: 

*“I prefer that my wife deliver our child at home because the Dai charges only 500–1000 rupees. If I were to take her to the hospital, I would have to spend close to 10,000 rupees for a routine delivery and more in case of any complications.”* (Man, 48 years, FGD)

Building on this point, another man stated: 

*“I will try and have the baby delivered at home and if there are any complications then I will rush my wife to the hospital.”* (Man, 40 years, FGD) 

In contrast to male respondents, women emphasized the importance of safeguarding the health of their child and showed a preference for institutional deliveries. According to one mother:

*“I would like to deliver all my children at the hospital. The medicine, injections and trained staff that are available at private health facilities are better than what we get from TBAs at home.”* (Woman, 33 years, FGD) 

Our research also indicated that many women are unable to exert their influence on this vital decision. For instance, despite wanting to deliver their babies at medical institutions, mothers face resistance from their husbands:

*“Many women confide in me that they would like to deliver their children at the private hospital, but they are not permitted. They ask me to speak to their husbands. Sometimes they listen but usually the man refuses.”* (LHW, 45 years, IDI)

Similarly, another woman reported:

*“I have had a really big argument with my mother-in-law and my husband over the delivery of our second child. I want to go to the private medical clinic, but they will not let me. They say that it is too expensive and ask me why I should get special treatment.”* (Woman, 40 years, FGD)

Despite having a strong desire to utilize health services, many women are prevented from doing so. This occurs through a combination of financial constraints and uneven power relations. 

#### 3.2.3. Restricted Women’s Mobility

Cultural rules that prohibit the interactions of unmarried men and women prevented pregnant women from visiting health facilities. As a result, many women had to delay visiting clinics because they lacked the presence of a male member from the family: 

*“I observed severe labor pain and started bleeding. At the time my husband was working in the fields and only came back home in the evening. I was taken to hospital by my husband because I cannot go alone to health facility without a male companion.”* (Woman, 21 years, FGD)

Similarly, our interviews with LHWs also revealed that ANC and PNC highly compromise due to the prevalence of this rule: 

*“We counsel women to seek timely ANC and PNC services, but women often miss essential ANC visits because they remain dependent on men to move out from the home. Sometimes, they [males] are busy in the field and women do not seek ANC at all.”* (LHW, 38 years, IDI)

These responses indicate that despite wanting to visit health facilities, many women were unable to do so. Rather than seek care when it is most needed, mothers had to wait for the presence of a male member so that they may receive the treatment they need.

### 3.3. Structural Barriers 

Structural barriers refer to the presence of macro-level factors such as policies, practices, and procedures that prevent people from accessing health services. In our study, we identified two such barriers: ineffective referral systems and prohibitively expensive transportation services.

#### 3.3.1. Ineffective Referral Systems

Our interviews revealed that patients were left frustrated with referral system present at public health facilities. Rather than deal with the frustration of navigating the referral system, patients would opt out of services altogether. The basic health unit located in close proximity to the village serves as the first point of contact with the health system. It is from the basic health unit that patients are referred to other health facilities based upon their needs. However, due to a lack of coordination between different levels of health facilities, patients are often left frustrated and eventually opt for home-based care. According to one mother:

*“When I went to the basic health unit, I was referred to the district hospital for ultrasound scan. We took four days to arrange money for my visit and when we reached the hospital, we were told that did not have a functional ultrasound machine.”* (Woman, 32 years, FGD)

Further highlighting this point, a man shared his experience:

*“Whenever the basic health unit refers us to a doctor, he is not present. We make arrangements for transportation and accommodation only to find out that these services are unavailable. At the end it is best to just opt for home care.”* (Man, 32 years, FGD)

To conclude, our interviews indicate that a lack of coordination between various levels of health facilities leads to a weak and ineffective referral system. This causes a highly frustrating experience for patients and eventually poor service utilization. 

#### 3.3.2. Prohibitively Expensive Transportation Services

Both men and women pointed out that they were unable to access care during pregnancy because of prohibitively expensive transportation services. While the basic health unit is located in close proximity to rural residents, patients are often referred to the district hospitals for scans and treatment. These can be located 20–100 km away from these villages. With no public transportation facilities present, patients must hire private transportation services, which are costly. This point was highlighted by a man who stated:

*“When I go to the field to sell my labor, they pay me Rs.400 for the day. If I have to transport my wife to the district hospital it will cost me Rs. 1500. These are nearly my wages for the whole week.”* (Man, 40 years, FGD)

In many cases, families have to sell off important assets in order to afford access to basic services. According to one woman:

*“When we found out that our child was positioned the wrong way, we knew that this would be a complicated delivery and that we would have to make many visits to the hospital. In anticipation my husband sold one of our goats so that we could have some many to make arrangements.”* (Woman, 32 years, FGD) 

The responses from these interviews indicate that despite the presence of clinics and government hospitals, pregnant women in Thatta are not able to access services due to expensive transportation services. 

## 4. Discussion

To our best knowledge, this is the first study that explores the multifaceted barriers that pregnant women face in utilizing maternal and newborn healthcare services in Thatta, Sindh. To this end, we used an interpretive methodology to identify the individual, sociocultural, and structural-level barriers that inhibit maternal and newborn health services utilization. The individual barriers unearthed by this study included mistrust towards public health facilities and inadequate symptom recognition. The identified sociocultural barriers were aversion to biomedical interventions, gendered imbalances in decision making, and women’s restricted mobility. Lastly, the identified structural barriers were prohibitively expensive transportation services and ineffective referral systems (Figure 1). 

### 4.1. Individual Barriers 

Our interviews reported that mothers have a trust deficit towards public health facilities. They cited the apathetic attitude of staff and poor service delivery as major reasons for avoiding these facilities. This finding is consistent with a national survey which reported that only 32.6% of patients attend public health facilities in Pakistan [23]. Different studies have also pointed out that private health facilities are preferred in Pakistan due to better facilities and quality of care [24,25,26]. Similarly, Mahrooj et. al. identifies deficiency in facility resources and the indifferent attitude and non-availability of the staff as factors that lead to poor antenatal uptake at public health facilities [6].

Our study also indicates that women in Thatta show inadequate symptom recognition. They are unable to detect the presence of pregnancy complications in their early stages. It is only when these symptoms reach acute proportions that they are galvanized into seeking care. Symptom recognition may be a product of cultural conditioning. In order for someone to be recognized as ill, he or she must be considered as ill in the home culture. Therefore, certain kind of bodily pains may not be recognized as signs of illness. For example, back pain is considered as an illness in Western medicine but is rarely seen as pathological amongst the world’s laborers [28]. 

Inadequate symptom recognition could also stem from constrains within the local environment. Studies from South Asia point towards several such constraints including the distance to health facilities, the societal negligence of women’s health, and their lack of decision-making power. While all these factors may be pervasive, our study indicates that the distance to the health facility may play a role in delayed symptom recognition. For example, the closest health facility to a village is the basic health unit [29]. Studies from rural Pakistan report that basic health units have limited facilities and are staffed by apathetic healthcare providers [30,31]. Therefore, in order to receive quality treatment, women either go to a private clinic or travel to the nearest city [30]. According to our interviews, both these choices are prohibitively expensive. Therefore, it is possible that women may downplay their symptoms to avoid a financial loss by seeking care only in acute situations. 

### 4.2. Sociocultural Barriers 

A large proportion of our respondents were apprehensive towards biomedical interventions. They reported that in comparison to local remedies, biomedical interventions cause negative effects. In particular, they were hesitant to avail micronutrients and vaccinations, citing that both may cause pregnancy complications. Vaccinations for neonatal tetanus are low in Pakistan and according to a recent provincial survey, only 56.1% of women were protected against neonatal tetanus in Sindh [32]. 

To understand this phenomenon, one must take into account that biomedicine is one of several concurrently running medical systems in the region. Biomedical explanations must compete with accounts coming from: Ayurvedic, Unani, and Indian medicinal influences [33]. In all medical systems other than biomedicine, illness arises from a misbalance in the body’s humoral fluids, which causes excessive heat or coldness within the body [34]. From the perspective of a medical practitioner, a vaccine or nutritional supplement is boosting the body’s immunity. However, from the perspective of a rural mother influenced by Ayurveda or Unani medicine, the same supplement could be misbalancing the body and producing excess heat. This can explain why some mothers associate micronutrients and vaccines with pregnancy complications.

In addition, our interviews revealed that male respondents preferred to deliver their children at home, whereas female respondents were inclined towards having deliveries at private medical institutions. However, in most cases it was the will of the male that prevailed as deliveries took place at home. This occurred because private health facilities are prohibitively expensive for rural families and males exerted their agency to push their wives towards delivering children at home. This finding is consistent with data from Pakistan which indicates that 41% of deliveries in rural areas take place at home [5]. Similarly, Mcnojia et al. also cite that husbands restrict wives from visiting doctors during pregnancy in Thatta, Sindh [35]. A reason for this phenomenon may stem from the fact that conventional gender norms in Pakistan dictate that men are responsible to financially provide for the family [36]. As a result, they hold the final authority on most economic decisions. Moreover, women do not seek care independently during pregnancy and childbirth due to their restricted mobility. In such circumstances, women remain dependent on male members to take them to hospital. Studies from Pakistan corroborate this and highlight that home births are highly prevalent because of difficulties in obtaining permission to visit a health facilities, financial dependency, distance to health facilities, and costly transportation [37]. Moreover, the subservient status of women keeps them disempowered and adds further constraints for timely maternal care [38].

### 4.3. Structural Barriers 

Our respondents indicated that transporting patients to the hospital was a significant barrier to health service utilization. This was because in all cases patients had to arrange private transportation for routine check-ups. In addition, essential equipment for pregnancy check-ups such as ultrasound machines are only present in district-level hospitals which can be located anywhere between 20 to 70 km from rural villages [35]. The average cost to arrange a private vehicle across this distance in Thatta is PKR 2000 rupees (USD 13 US) for a round trip. Given that the average daily income for a household is PKR 500 (USD 3), this cost is likely too much to bear. 

A poor referral system serves as a further disincentive to service utilization. As indicated above, arranging transportation is an expensive and difficult task for most rural households. The first institutional contact for an individual living in a rural area is the basic health unit from where expecting mothers are referred to a larger facility such as a hospital for more sophisticated treatment [39]. Many of these facilities are located far away and require costly transportation arrangements. Our research indicates that despite making these difficult arrangements, patients are not able to receive the care they need. This occurs because of numerous gaps within the referral system. For example, our interviews revealed that patients were not able to complete their referral visits due to defunct equipment, absent doctors, and unplanned facility closures. Better coordination between different levels of institutions within the referral system can ensure that this information can be shared with patients beforehand. This would prevent the wastage of resources and would ensure that patients are not discouraged from seeking care at public health institutions.

Our qualitative data may have limitations due to the fact that participants were purposively selected, and their experiences may not be uniform across all rural areas of Pakistan. Another limitation in our study is that we did not interview mothers-in-law, whose role and perspective is pivotal in decision making during pregnancy and childbirth. It is also important to highlight that we only collected data through FGDs from men and women and not through in-depth interviews. This may have prevented us from obtaining certain sensitive information. 

## 5. Conclusions 

The rural district of Thatta located in Sindh, Pakistan, has the worst maternal and newborn health indicators across all LMICs. Moreover, this phenomenon is occurring at a time where more resources are being put towards increasing health services in the area. Research indicates that maternal–newborn health is highly influenced by multifaceted barriers. However, to date no study has provided an in-depth understanding of what prevents health service utilization in Thatta. Using an interpretive research methodology based on in-depth interviews and focus group discussions, we unearthed seven barriers that were individual, sociocultural, and structural in nature. 

The individual barriers identified in this study can be addressed through a combination of capacity development of service providers, health education, and community engagement sessions to women in reproductive age and mothers-in-laws. For example, a training program emphasizing empathetic care can address the trust deficit many patients associate with public health facilities. At the same time, the content of health education sessions can be adjusted according to the findings of this study so that mothers can have better symptom recognition and more autonomy over health-related decisions. Moreover, these sessions could also be used to address the sociocultural barriers identified in this study. It is also important to organize community engagement sessions at the village level to facilitate balanced decision making between males and females. An emphasis can also be placed on highlighting the benefits of supplements and vaccinations along with addressing the negative beliefs associated with their consumption. Lastly, a well-coordinated referral system and a network of subsidized ambulance services would address the structural barriers of this study and put public health facilities within the reach of patients.

## Figures and Tables

**Figure 1 healthcare-09-01314-f001:**
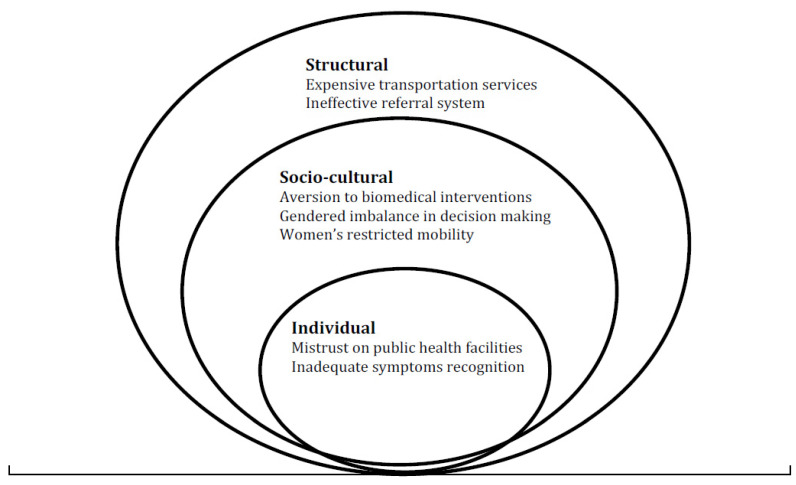
Socioecological model highlighting major health seeking barriers identified in this study.

**Table 1 healthcare-09-01314-t001:** Study participants (*n* = 60).

Stakeholders	Total Interviews ^1^	Total FGD ^2^ Participants	Total Participants
Women	-	28 (4 sessions)	28
Men	-	26 (4 sessions)	26
LHW ^3^	6	-	6
Total	6	54 (8 sessions)	60

^1^ In-depth Interviews; ^2^ Focus Group Discussions; ^3^ Lady Health Workers.

**Table 2 healthcare-09-01314-t002:** Background characteristics of healthcare workers (*n* = 6).

Code	Type of Healthcare Providers	Education	Type of Employment	Age in Years	Working Experience in Years
1	LHW	Matric	Public sector	48	19
2	LHW	Middle	Public sector	45	19
3	LHW	Matric	Public sector	39	15
4	LHW	Intermediate	Public sector	54	24
5	LHW	Matric	Public sector	33	12
6	LHW	Matric	Public sector	37	14

**Table 3 healthcare-09-01314-t003:** Background characteristics of community participants (*n* = 54).

Sociodemographic Variables	Categories	f (%)
Gender	Male	28 (51.9%)
	Female	26 (48.1%)
Age	15–24 years	14 (25.9%)
	25–34 years	19 (35.2%)
	35 and above	21 (38.9%)
Education	Illiterate	32 (59.3%)
	Primary	13 (24.0%)
	Middle and above	9 (16.7%)
Number of children	2–3	23 (42.6%)
	4–5	19 (35.2%)
	6 and more	12 (22.2%)

## Data Availability

The datasets used and/or analyzed during the current study are available from the corresponding author on reasonable request.

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
