# Peer review of "We Won’t Go There: Barriers to Accessing Maternal and Newborn Care in District Thatta, Pakistan"

_healthcare, 2021, doi:10.3390/healthcare9101314_

Round 1
Reviewer 1 Report
The study is a reasonably designed one, which is likely of relevance in the Pakistan health system or in other low-middle income countries where women's access to adequate antenatal and postnatal care is often reduced. Some further attention to detail of the presentation of the results with consistent use of formatting design, superscript referencing and quotation/participant identifiers requires attention. There were several small either typographical or incorrect words usage, even though the article is generally well written. The conclusion needs to be more realistic. It would take significant time and targeted educational initiatives to address the cultural conditions in which women's views are secondary to their husbands and thus for them to move towards having more autonomy over their health-related decisions. Suggesting small step innovations at the village and basic health unit level would start this process. Overall, the paper is publishable once some attention to the comments and highlights are made and contributes to establishing the evidence base required to make change.

Author Response
Healthcare (ISSN 2227-9032)
Manuscript ID: healthcare-1369158
We won’t go there: Barriers to accessing maternal and newborn care in district Thatta, Pakistan
Authors
Muhammad Asim, Sarah Saleem, Zarak Husain Ahmed, Imran Naeem, Farina Abrejo, Zafar Fatmi, Sameen Siddiqi
Comments and Suggestions for Authors
REVIEWER# 1
Reviewer’s comment:
The study is a reasonably designed one, which is likely of relevance in the Pakistan health system or in other low-middle income countries where women's access to adequate antenatal and postnatal care is often reduced. Some further attention to detail of the presentation of the results with consistent use of formatting design, superscript referencing and quotation/participant identifiers requires attention.
Author’s reply:
Thank you so much for your kind words.
Reviewer’s comment:
There were several small either typographical or incorrect words usage, even though the article is generally well written.
Author’s reply:
We have carefully reviewed again and now we believe that we have addressed typographical and spell-related issues.
Reviewer’s comment:
The conclusion needs to be more realistic. It would take significant time and targeted educational initiatives to address the cultural conditions in which women's views are secondary to their husbands and thus for them to move towards having more autonomy over their health-related decisions.
Author’s reply:
Thank you for highlighting this point. We do agree with this suggestion. We have written a sentence as suggested in the conclusion section.
“It is also important to organize community engagement sessions on women autonomy over their health-related decisions and to approach health facilities in case of any emergency.”
Reviewer’s comment:
Suggesting small step innovations at the village and basic health unit level would start this process.
Author’s reply:
We have mentioned that health education sessions and community engagement sessions at the village/ community level should organize for behavior change.
“It is also important to organize community engagement sessions at village level on women autonomy over their health-related decisions and to approach health facilities in case of any emergency. An emphasis can also be placed on highlighting the benefits of supplements and vaccinations and negative beliefs associated with their consumption can be clarified through health education and community engagement sessions at village level.”
Reviewer’s comment:
Overall, the paper is publishable once some attention to the comments and highlights are made and contributes to establishing the evidence base required to make change
Author’s reply:
Thank you so much!

Reviewer 2 Report
General remarks
The paper deals with a fascinating topic— the multifaceted barriers to health service utilization in rural Pakistan—and this is a relatively blind spot in the scholarly literature. That said, a few things need tightening though to further enhance the quality of the paper.
Introduction
- The paper is comprehensively referenced to extant scholarship on the topic and the authors clearly identify a research gap which they attempt to address.
-The authors should insert a paragraph after line 72 that states how the paper is organized, as this will guide readers. For instance, ‘the rest of the paper is divided as follows: the next section outlines our research methodology, then we present our results…’ In other words, this should be the last paragraph before they move to the next section (that is, ‘Methods’).
Methods
- I am surprised that the authors relied exclusively on focus group discussions to gather data on non-healthcare research participants. The focus group discussions should have been complemented by in-depth interviews with women and men with children under the age of two. It is well known that people respond differently to questions asked in groups and those asked in a one-on-one setting. The point that I am emphasizing here is that the authors missed out on very rich data, as they did not conduct in-depth interviews especially with women with children under the age of two.
Discussion
-The authors wrote: “The main strengthen of this study to interview men, women and LHWs to produce credible, authentic, and transferable findings that reflect the transparent picture of multifaceted barriers of seeking maternal and newborn care in Thatta” (Lines 435-438). This is misleading, as the authors had focus group discussions (FGDs) with a group of men and women, as outlined in their “Methods” section. So, this should be rephrased, and the authors should be transparent by saying they held focus group discussions with women and men. From a research methodology perspective, there is a clear difference between one-on-one in-depth interviews and focus group discussions.
-Also, the fact that the authors did not conduct in-depth interviews with women and men with children under two is a major limitation which must be mentioned in the limitations paragraph.
Author Response
Healthcare (ISSN 2227-9032)
Manuscript ID: healthcare-1369158
We won’t go there: Barriers to accessing maternal and newborn care in district Thatta, Pakistan
Authors
Muhammad Asim, Sarah Saleem, Zarak Husain Ahmed, Imran Naeem, Farina Abrejo, Zafar Fatmi, Sameen Siddiqi
Comments and Suggestions for Authors
REVIEWER’S 2
The paper deals with a fascinating topic— the multifaceted barriers to health service utilization in rural Pakistan—and this are a relatively blind spot in the scholarly literature. That said, a few things need tightening though to further enhance the quality of the paper.
Reviewer’s comment
Introduction
- The paper is comprehensively referenced to extant scholarship on the topic and the authors clearly identify a research gap which they attempt to address.
Author’s reply:
Thank you so much!
Reviewer’s comment
-The authors should insert a paragraph after line 72 that states how the paper is organized, as this will guide readers. For instance, ‘the rest of the paper is divided as follows: the next section outlines our research methodology, then we present our results…’ In other words, this should be the last paragraph before they move to the next section (that is, ‘Methods’).
Author’s reply:
Thank you so much for the comment. The scientific papers follow the similar patterns. For example, the research paper is comprised on the introduction, methods, results, discussion and conclusion section. We also followed this pattern in this paper. If we will write these lines, of course, they will not add any value in this article. This information is obvious for the readers that are the part of each research article.
Reviewer’s comment
Methods
- I am surprised that the authors relied exclusively on focus group discussions to gather data on non-healthcare research participants. The focus group discussions should have been complemented by in-depth interviews with women and men with children under the age of two. It is well known that people respond differently to questions asked in groups and those asked in a one-on-one setting. The point that I am emphasizing here is that the authors missed out on very rich data, as they did not conduct in-depth interviews especially with women with children under the age of two.
Author’s reply:
Thank you so much for highlighting for not collecting in-depth interviews from women and men. We collected data through FGD from women and men that give the opportunity to generate a holistic picture of common people’s interests. We did not collect in-depth interviews since our study did not carry any sensitive or personal information that people were hesitant to express in the FGDs. However, we have mentioned this as a limitation in our study in the last para of the discussion section.
Reviewer’s comment
Discussion
-The authors wrote: “The main strength of this study is to interview men, women and LHWs to produce credible, authentic, and transferable findings that reflect the transparent picture of multifaceted barriers of seeking maternal and newborn care in Thatta” (Lines 435-438). This is misleading, as the authors had focus group discussions (FGDs) with a group of men and women, as outlined in their “Methods” section. So, this should be rephrased, and the authors should be transparent by saying they held focus group discussions with women and men. From a research methodology perspective, there is a clear difference between one-on-one in-depth interviews and focus group discussions.
Authors reply:
Thank you for highlighting this important point. We have removed this sentence from our manuscript. We do agree that the recommendation of the reviewer.
Reviewer’s comment
-Also, the fact that the authors did not conduct in-depth interviews with women and men with children under two is a major limitation that must be mentioned in the limitations paragraph.
Author’s reply:
We have mentioned this as a limitation in this study in the study limitation paragraph.
“It is also important to highlight that we only collected data through FGDs from men and women that as another limitation in our study since we did not collect data through in-depth interviews”.

Round 2
Reviewer 2 Report
The authors have addressed my concerns.